# *"Unanswered questions"*: Acceptability of a personalised breast cancer screening strategy in lower-risk women by healthcare professionals in the context of the MyPeBS study

**Celmira Laza-Vásquez**[1], **Marta Román**[2☯*], **Suzette Delaloge**[3],
**Sandrine De Montgolfier**[4], **Xavier Castells**[2☯*], on behalf of the MyPeBS Executive Committee¶

**1** Department of Nursing and Physiotherapy, University of Lleida, Lleida, Spain, **2** Department of Epidemiology and Evaluation, Hospital del Mar Research Institute, Hospital del Mar, Barcelona, Spain, **3** Department of Cancer Medicine, Institut Gustave Roussy, Villejuif, France, **4** Inserm, IRD, SESSTIM, ISSPAM, Aix Marseille Univ, Marseille, France

¶ The complete membership of the author group can be found in the Acknowledgments.
☯ These authors contributed equally to this work.
* mroman@hmar.cat (MR); xcastells@hmar.cat (XC)

## Abstract

### Background

Based on the relevant differences in breast cancer risk, strategies for personalizing population screening have been proposed. The present study was conducted in the context of the MyPEBS study, which seeks to improve the evidence on the benefits and risks of implementing personalized screening. We explored the views of healthcare professionals involved in the MyPeBS study on the acceptability to extend the screening interval in women at lower risk, as proposed by MyPeBS.

### Methods

Qualitative interpretative descriptive study. Twelve health professionals were interviewed individually and in a discussion group. The transcripts were analysed using thematic analysis.

### Results

For healthcare professionals, the low risk estimate was good news for women. However, the acceptability of decreasing screening frequency was not homogeneous. A lower uptake seemed to be mainly influenced by previous participation in current population-based screening. Several uncertainties were raised for acceptability: the benefits of such personalization for lower-risk women, how to guarantee equity, and the feasibility of this de-escalation in terms of human and financial resources.

**Data availability statement:** Data consists of recorded group discussions and the corresponding transcriptions. Languages used were Catalan and Spanish. Data cannot be shared publicly because interviews were confidential and the transcriptions contain personal sensitive data. Data may be available upon request from the Ethics Committee at Hospital del Mar Research Institute for researchers who meet the criteria for access to confidential data (contact via: CEIC-PSMAR@imim.es; mroman@hmar.cat).

**Funding:** The MyPeBS project is funded by the European Union's Horizon 2020 research and innovation programme under grant agreement N° 755394. MyPeBS has received funding from project PI19/00007 of the Instituto de Salud Carlos III (ISCIII) and co-funded by the European Union, for the conduct of the study in Spain. The funders played no role in the study design, data collection and analysis, manuscript drafting, or the decision to publish the results. The funding information (funders who received the grants). Instituto de Salud Carlos, III, PI19/00007, PhD Marta Román, PhD Xavier Castells, H2020 European Research Council, N° 755394, PhD Marta Román, PhD Suzette Delaloge, PhD Sandrine De Montgolfier.

**Competing interests:** The authors have declared that no competing interests exist.

## Conclusion

Previous participation in screening was the most significant factor for women's low acceptance of decreasing screening frequency. Future research should refine the views of health professionals and women. Recommendations should be made to promote participation in personalized screening of women at lower risk.

## 1. Introduction

The risk of developing breast cancer over a woman's lifetime depends on characteristics such as age, breast density, family history of breast cancer, history of benign lesions, or genetic predisposition, among other factors. The difference in risk can be substantial between two women based on these characteristics. However, population-based breast cancer screening is based solely on age. It is in this context that it has been proposed to evaluate the effectiveness of screening based on a woman's individual risk [1].

There is still no evidence to suggest that personalized breast cancer screening is no worse than or potentially better than current screening in reducing breast cancer mortality [2]. Specifically, there is insufficient evidence to demonstrate that the primary goal of screening, which is to detect the killing cancers early, can be improved for all women with a personalized model that screens higher risk women more frequently and lower risk less frequently. However, it is suggested that this strategy could increase the benefits and reduce the harms of current screening overall. Two prospective multicenter randomized controlled trials are being developed internationally to generate evidence on the effectiveness of personalized screening in reducing breast cancer-specific mortality compared to current screening (Wisdom in the US and MyPeBS in Europe and Israel) [2,3].

Stratified breast cancer screening estimates individual breast cancer risk by using various risk factors and proposes recommendations (screening interval, start age, and type of tests) to be tailored to different risk groups [4]. Several models are available for predicting and stratifying risk and generally include age, family history of breast cancer, previous benign breast conditions, breast density, and polygenic risk scores [5].

Both MyPEBS and WISDOM use three main levels for risk stratification: higher, average or medium, and lower risk, based on the probability of developing cancer in the next five years. In MyPeBS for the highest risk group, a mammogram (or MRI for dense breasts) every year is recommended for the higher risk group, a 2D mammogram (or tomosynthesis for dense breasts) every two years is recommended for the average risk group, and a 2D mammogram every four years for women at lower risk than the average. This group of women is particularly relevant as a longer screening interval can improve specificity but does not guarantee the same sensitivity. Overall, it is expected that sensitivity will be increased in the highest-risk group without significantly compromising specificity (false positives), and specificity will be increased in the lowest-risk group without significantly compromising sensitivity, particularly in the detection of early-stage breast cancers. Based on the values and expectations of

women, healthcare professionals and other stakeholders, compromising the sensitivity of screening is of greater importance than compromising specificity.

In addition, personalisation increases complexity due to the socialisation of the message and its communication to women, which makes the role of healthcare professionals essential in explaining complex concepts such as relative and absolute risk reduction, overdiagnosis and false positive results. Furthermore, complexity increases for future implementation of personalized screening, as it will increase the workload of professionals, it will require additional staff training and poses challenges for extending the intervals between screening tests [3]. These challenges include one of the most complex issues in implementation: the potentially lesser acceptance of extending screening intervals among women estimated to be at lower risk [4,6]. In turn, this low acceptance seems to involve various factors, including women themselves (their beliefs, fears, knowledge, attitudes, experiential knowledge), the healthcare professionals and system's organisation, and the personalised strategy itself [7]. All these considerations could hinder the transition to a new screening strategy, as personalisation must be acceptable to different stakeholders involved from the clinical, social, and ethical points of view [8].

Healthcare professionals play a determining role in the implementation of the new screening strategy, as their perceptions may influence women's acceptability and decision to participate; therefore, getting closer to their opinions regarding this screening approach will contribute to the effective of the implementation [9–12].

There is limited evidence on the opinions of healthcare professionals involved in personalised screening projects regarding the acceptability of this strategy. In general, opinions are positive, considering it a logical step in breast screening. They believe that this strategy will particularly benefit high-risk women by increasing the frequency of screening, reducing mortality from the disease and increasing the survival of women diagnosed. For lower-risk women, they report a reduction in overdiagnosis and false positives. These advantages will increase the efficiency and effectiveness of the use of healthcare system resources. The main disadvantages highlighted are the potential resistance of lower-risk women to the recommendation to reduce the intensity of mammographic screening [13,14].

Only the BC-Predic project in the United Kingdom has thoroughly investigated the views of those involved on reducing screening intensity. Health care professionals consider this logical, but they are cautious in recommending it due to doubts about the accuracy and stability of risk estimates and the increase in interval cancers, as well as a misperception among women that lower risk is synonymous with no risk [15].

The aim of this study was to explore the views of healthcare professionals involved in the MyPeBS trial on the acceptability of a personalised strategy for breast cancer screening in women deemed at lower risk.

## 2. Materials and methods

### 2.1 Design

We performed a descriptive, interpretative, qualitative study.

### 2.2 Participants

Healthcare professionals who were part of the MyPeBS (International Randomized Study Comparing personalized, Risk-Stratified to Standard Breast Cancer Screening In Women Aged 40–70) in Spain between 2021 and 2023 and agreed to participate in the study.

MyPeBS is an ongoing international randomised, open, multicenter study initiated in 2019 and funded by the European Union's Horizon 2020 Research and Innovation Programme. Its main objective is to demonstrate the non-inferiority of personalised screening in reducing the incidence of stage 2 and higher breast cancer (2+) compared with standard screening, according to the current national guidelines in each participating country [2]. Recruitment was completed in mid-2023, with 53,143 women aged between 40 and 70 recruited in Belgium, France, Israel, Italy, Spain and the United Kingdom. In Spain, 3,000 women were recruited. In Spain, recruitment took place at the Hospital del Mar and the Hospital de la Esperanza, which are part of the Parc de Salut Mar.

Fourteen healthcare professionals participating in the MyPeBS project in Spain were invited in person and/or by email, and three did not respond. The sample included 11 professionals with an average of 15.5 years of experience in the breast cancer screening programme (range 2–28 years), with different professions and roles within the project: three doctors specialising in radiology and one in gynaecology and obstetrics; one doctor specialising in public health and epidemiology; four senior diagnostic imaging technicians; and two research technicians responsible for recruiting women for the study. 73.7% of the subjects were women and the average age was 40.7 years (range 25–65 years).

Data collection and analysis were conducted at the same time and from informant number 8 onwards, no new information was obtained that would permit further expansion or deepening of the results. This finding suggests that theoretical saturation of information was achieved.

## 2.3  Data collection and analysis

Participants were interviewed, using a semi-structured interview guide, at Hospital del Mar by a researcher with broad experience in qualitative research. Initially, a focus group was conducted with four senior diagnostic imaging technicians. The remaining participants were individually interviewed once, with both data collection techniques lasting between 40 minutes and 1.5 hours. The interview script included the following questions:

- How do women perceive personalised breast cancer screening?

- How do women perceive having a lower risk of breast cancer?

- What factors in women influence their acceptance of extending screening intervals?

- What are the recommendations to encourage acceptance of the recommendation to extend screening intervals?

All interviews were audio-recorded and transcribed verbatim by a company contracted for this purpose. The data were analysed using a thematic analysis [16].

After reading and re-reading the transcripts to become familiar with the data, a researcher conducted the initial coding, identifying relevant data. Subsequently, the resulting codes were discussed by all the researchers, and themes and subthemes were defined until the body of results was constructed.

## 2.4  Reflexivity of the research team

The research team, composed of researchers from different health-related professions with broad experience in screening, maintained a reflexive stance, periodically discussing progress and difficulties, and making critical decisions. Field notes from the interviews were taken to deepen understanding of the research objective [17] and were considered during data analysis. To ensure the credibility of the results, information from various participant profiles and roles was triangulated. The final results were returned to participants via email and their suggestions were incorporated into the final results.

## 2.5  Ethical considerations

Ethics approval was obtained in each participating country. In Spain, approval to conduct the MyPeBS was granted by the Ethics Committee at Hospital del Mar Research Institute (20198666-I). For this qualitative study, approval was obtained from the Research Ethics Committee for Medicinal Products of the Parc de Salut Mar (Act dated 16/01/2024). Individual written informed consent was obtained from each participant.

## 3.  Results

### 3.1  Theme 1. Views on personalised breast cancer screening

**3.1.1  Positive views of personalised screening.**  Healthcare professionals viewed personalisation positively, seeing it as beneficial for women. They believed it was needed to improve screening and, like other fields of personalised medicine (diagnosis and treatment), that it represents the future of early breast cancer detection.

They believed that women also viewed personalization positively, as it adds value to screening by incorporating new factors for early detection (genetic testing and breast density). It also provides women with previously unknown information, especially those who believe to be at high risk (those with a family history and dense breasts). Furthermore, it involves individualized care for each woman, through investigation of the factors influencing risk estimation and follow-up over time. Thus, to their opinion, women believed that this strategy provided more detailed care and that there was greater concern for their health.

**3.1.2  Acceptability of personalization of breast cancer screening in the MyPeBS study.**  Healthcare professionals explained that women accepting participation in the MyPeBS study showed a certain resistance to a change of screening strategy towards a risk-based screening, partly due to the pervasive influence of the current model perpetuated by the media and the healthcare system over the years. Consequently, they believed that biennial mammography had become deeply ingrained, with a prevalent *"warning"* narrative surrounding breast cancer and its risks. However, they thought that women's acceptance of a new risk-based approach was not uniform and was mainly influenced by two factors: risk perception and previous participation in population-based and/or opportunistic screening. High acceptability was associated with having a family history of breast cancer and benign lesions, as well as with not having started participation in current screening and/or regular opportunistic screening.

Healthcare professionals also felt that acceptance of this new general approach was lower among women who believed themselves to be at lower risk of breast cancer, especially those without a family history of breast cancer or other cancers, and those who underwent biennial mammograms in population-based screening and/or annual screening in the private healthcare system (S1 Table).

### 3.2 Theme 2. Views on extending screening intervals among women at lower risk

Healthcare professionals believed that women classified as lower risk often interpreted this information as good news, providing them with a sense of reassurance, but they also highlighted the risk of creating a *"false sense of security"* when lower risk was mistaken for no risk at all. However, they believed that acceptance of the recommendation to extend screening intervals was lesser due to three main factors:

**Mistrust.** Professionals explained that women tended to perceive the recommendation as a cost-saving measure driven by public healthcare. Moreover, their apprehension about developing advanced-stage disease during the 4-year interval between mammograms stemmed from personal experiences and anecdotes from other women in their circles who had developed breast cancer despite participating in screening programmes. This strengthened their belief that the disease could strike at any time.

**Previous screening participation.** According to healthcare staff, prior participation was a differentiating factor that instilled a sense of *"security"* in women due to the regularity of biennial mammograms. Some women even considered the 2-year interval too lengthy, prompting them to alternate between biennial mammograms and opportunistic screening in public and/or private healthcare. Consequently, lower-risk women tended to feel that personalisation *"took away"* rather than "*added*" to early breast cancer detection.

**Socioeconomic characteristics.** Professionals believed that socioeconomic status was linked to less acceptance of personalisation in two ways: firstly, women with lower socioeconomic, educational, and health literacy levels struggled to comprehend the recommendation to expand screening intervals. In contrast, reluctance to accept personalisation among women with medium and high socioeconomic status stemmed from their past experiences of regular opportunistic screening and their access to private healthcare due to their financial means. Consequently, they were more inclined to accept the suggestion of *"annual check-ups"* that included mammograms and other tests (breast ultrasound and cervical cytology) (S2 Table).

### 3.3  Theme 3: Healthcare professionals "doubts" about the implementation of personalized breast cancer

While healthcare professionals generally viewed personalisation positively, they harboured several *"doubts"* about its implementation.

**Benefits of personalisation in lower-risk women.** In the absence of robust evidence of the benefits of personalised breast cancer screening compared to the current screening model, healthcare professionals were sceptical of the recommendation to extend intervals in lower-risk women. Consequently, they stressed the need for greater certainty about the sensitivity and specificity of risk estimation and the imperative of ensuring that risk stratification and mammography intervals were optimal. They also required evidence that personalisation reduces the harms of current screening practice, such as false positives, interval cancers, and overdiagnosis.

Participants believed that having robust evidence would give them the certainty and confidence needed to recommend the expansion of screening intervals, providing reassurance to both themselves and women.

**Risk communication.** Risk communication was seen by healthcare professionals as complex due to women's difficulty in understanding and interpreting this concept, as well as their preconceived notions about *"risk"*. There was also uncertainty about the most appropriate way to communicate risk, as understanding and interpretation of lower risk varied and could lead to misconceptions, fear, and confusion.

**Concerns about women's participation.** Healthcare professionals voiced several concerns about the best way to ensure equitable participation if personalized screening is introduced among all women, given the complexity of grasping personalisation and the potential uncertainty it might create. They predicted that some women might not fully understand it, while others, even if they did, might decide against participating.

**Feasibility of implementation.** Implementing personalised screening is a complex endeavour as it involves more than simply *"adapting"* the current model and requires *"designing"* a new one, which poses challenges in all domains. Consequently, healthcare professionals raised several doubts about the feasibility of its implementation in Spain.

In terms of human resources, health staff stressed the need for training in personalised screening and communication skills, especially for primary healthcare professionals. This training would enable professionals to effectively explain the personalisation process to women and ensure their understanding. Without such training, there was a risk of women's participation becoming a passive decision rather than a fully-informed active choice.

Participants also questioned whether the national health system had the necessary resources and capabilities to store and safeguard genetic information, as well as sufficient laboratories and staff to analyse a large volume of samples. Additionally, they wondered about the feasibility of implementing an information system accessible to all healthcare professionals in both the public and private sectors, enabling them to calculate risk and access updated information. Doubts were also expressed about whether the health system could afford the initial economic and human costs of estimating risk for all women, conducting consultations with healthcare professionals, and managing any modifications to the initial risk.

The message of *"essential surveillance"*. Participants highlighted that private insurers advise women to undergo annual mammograms, without clear criteria for this decision and disregarding the guidelines of population-based screening. They encourage annual mammograms more as *"essential surveillance"* than as a screening strategy. Although most women do not have the financial means to pay for private healthcare, this recommendation would convey two messages that contradict those of the publicly-funded health system about personalised screening: an annual mammogram is necessary for early detection, and expanding screening intervals is not based on scientific evidence or improving screening but rather on the public healthcare system's interest in cutting costs (S3 Table).

## 4. Discussion

We summarised healthcare professionals' views on the acceptability of a personalised breast cancer screening strategy in lower-risk women. Our results are consistent with previous findings indicating a high acceptability of personalised breast cancer screening, which decreases with the recommendation to widen screening intervals [10,18]. However, it also offers novel and different perspectives on this issue to those already reported.

This study was conducted in the context of the MyPeBS study, which aims to improve the evidence on the effectiveness of a personalised screening strategy based on individual risk by increasing sensitivity in women at higher risk without

compromising the diagnosis of early stage breast cancers in women at lower risk. The aim of this study was to analyse the opinion of healthcare professionals, assuming that this hypothesis is met, but with the uncertainty that it has not been confirmed.

In this sense, healthcare professionals in our study described *"uncertainties"* about the future implementation of a lower-risk pathway. Perhaps the most compelling doubt, and not unique to our study, is the need for stronger evidence of the benefits and safety of personalisation in lower-risk women compared with current screening [15], which is the main goal of MyPeBS trial [2].

This uncertainty makes them *"wary"* of recommending less frequent screening for fear of an increase in interval cancers, and consequently reduce sensitivity [15]. In this situation, Australian healthcare professionals have questioned whether the benefits of personalised screening would outweigh the harms, and whether its implementation could be justified without strong evidence and clear guidelines [19].

Participating healthcare professionals identified women's previous participation in population-based and/or opportunistic screening as the most influential factor in this lesser acceptability, which has not previously been documented in studies exploring the acceptability of a lower-risk pathway. This, in turn, relates to another factor that emerged in the study: the safety that biennial mammography screening engenders for women.

This desire for *"safety"* has become one of the most documented factors in the resistance to the introduction of new screening strategies [6]; confronting women with a choice between the *"safety"* versus *"advantages"* of personalisation (possibility of individualised early detection and a better balance of benefits and harms) [15,20]. For this reason, several authors have proposed that women choose the screening modality as a way of offering them peace of mind [14] and to increase the acceptability of personalised screening [6].

In turn, the "*safety"* provided by biennial mammography is enhanced by another factor identified in our study that has not been previously documented: the message of *"essential surveillance"* through annual mammography screening that is indirectly disseminated by private healthcare through women who have access to these services. This message is problematic because it becomes part of the construction of women's *"experiential knowledge"* about breast cancer and screening [21]. It contradicts the public health message, jeopardises women's participation in screening and their relationship with their primary care physicians [22], thus hindering the effective implementation of personalised screening [18]. This also includes fear of breast cancer, which continues to shape responses to the disease and participation in screening. Two European studies suggest that despite a high level of knowledge about the disease fear levels are high, prompting them to undergo more frequent screening, or even seeking for early detection when population based screening is not offered in women younger than 50 years [23,24]. This fear has been amplified by the media, which emphasises the benefits of screening rather than its harms and leads to participation driven more by emotional than rational factors [25].

Other *"uncertainties"* about implementation highlighted by participants were the optimal way to communicate lower risk. They consider this key to women's understanding and informed decision-making. This challenge has been reported in other studies, highlighting the complexity faced by health professionals when conveying lower risk. Often, lower risk is perceived as no risk, making breast cancer as an irrelevant concern [10,26]. Finally, they drew attention to the fact that risk communication requires more time than that allowed in typical consultation and the availability of qualified health professionals [11].

Our results are also consistent with other reported concerns, such as women's willingness to provide personal and genetic information [13], ensuring equal participation of all women, poor training of professionals in personalised screening and elements of genetics [27,9], and the shortcomings of public health systems, which may have a negative impact on the feasibility of implementation in terms of human, financial, and structural resources [15,28].

If the findings from MyPeBS support wider screening intervals for lower-risk women, we need to promote acceptability among these women with the use of various tools and the adaptation of styles and formats to the levels of health literacy to provide information on personalisation and communicate risk [18,26]. Ensuring that women understand information

on personalisation will key to ensuring equitable access for all women [6,8]. This is especially relevant in a multi-ethnic setting such as Catalonia, where large migratory processes contribute to the diversity of languages, cultures, religions and levels of health and digital literacy; these factors may affect women's understanding of estimated risk and screening recommendations.

However, in addition to adequate risk information and communication, it is also necessary to develop strategies to provide follow-up and *"support"* to lower-risk women during the intervals between mammograms. Thus, face-to-face meetings with health professionals, reminder letters, helplines and websites are strategies proposed for this purpose and are consistent with the results of other studies [15].

Another essential and pressing issue raised in the study is the need to train primary care professionals, given the decisive role they will play in implementing a personalised screening model [29]. Two large studies reported that these professionals lack sufficient knowledge and need further training [30,31]. Dunlop et al [32] emphasised that training them to carry out a change in practice would be a challenge that involves taking into account their diverse activities and roles, and time constraints.

On the controversial issue of perceiving the decrease in screening frequency as a cost-cutting exercise by the public health system, it is essential to recognise and explicitly state that this is partly due to *"reallocation of resources"* for high-risk women, but that its primary motivation is safety and harm reduction in screening tests [14]. The above is closely linked to the necessary collaboration between the healthcare system, the media and patient organisations to create information and educational initiatives to facilitate the understanding of personalised screening and the extension of screening intervals in lower-risk women [33].

In order to inform, educate and socialise women about personalised screening strategies, community initiatives such as patient associations and the *"expert patient"* strategy could be beneficial, as they have proven to be valuable sources of information and support and promote a 'sense of community' among women [34,35]. Although little reported in the literature, these community-based initiatives are crucial to reduce disparities in breast cancer screening by addressing the challenges and barriers faced by underserved populations. By focusing on the local setting, these initiatives interact with their communities and tailor interventions to their needs, building trust and awareness, as well as increasing the uptake of screening services [36]. In addition, working with community leaders to tailor health messages would help promote the participation of women from ethnic minorities [26].

Emerging evidence supports the feasibility of risk-based breast cancer screening. The WISDOM Study has demonstrated that personalized screening can effectively stratify women according to their risk while remaining safe and acceptable to participants [3]. These findings provide essential support for the transition from age-based to risk-stratified screening strategies, although further refinement of risk prediction models and risk-reducing recommendations as well as effective risk communication to patients and health care professionals are still needed for large-scale implementation. Importantly, reducing the frequency of screening in women at lower risk needs to be monitored to avoid losing sensitivity in the detection of early stage breast cancers.

## 4.1 Strengths and limitations

To our knowledge, this is the first Spanish study that summarises the opinions and views of health professionals on the acceptability of a personalised screening strategy in lower-risk women in real-life implementation. Therefore, the results may be useful for decision-makers in the design and implementation of personalised breast cancer screening programmes in public health systems. The results are based on a diverse sample of participants with different professional profiles, screening experiences and roles within the MyPeBS project, and ages and genders. Furthermore, these insights were obtained in a real-life personalised screening application scenario. A limitation may have been one social desirability bias in the group discussion [37], which was mitigated by triangulation of information from other participants with diverse professional profiles.

## 5. Conclusions

Although our results are largely consistent with other previously published studies, they offer novel and relevant insights into the factors influencing the acceptability of women deemed to be at lower risk of breast cancer and recommendations for increasing the acceptability of personalisation. Future research should further examine the recommendations made and assess their feasibility. In addition, studies are needed to investigate the views of primary care professionals and women on extending screening intervals, and to propose strategies to promote acceptability for women deemed to be at lower risk.

## Supporting information

**S1 File. Standards for Reporting Qualitative Research (SRQR).**
(DOCX)

**S1 Table. Views on personalised breast cancer screening.**
(DOCX)

**S2 Table. Views on extending screening intervals among women at low risk.**
(DOCX)

**S3 Table. Healthcare professionals "doubts" about the implementation of personalized breast cancer.**
(DOCX)

## Acknowledgments

The authors acknowledge the healthcare professionals who participated in this study. They also acknowledge the MyPeBS Executive Committee, listed here in alphabetical order: Jean-Benoît Burrion, Suzette Delaloge, Sandrine De Montgolfier, Fiona Gilbert, Livia Giordano, Paolo Giorgi-Rossi, Michal Guindy, Harry De Koning, Cecile Vissac-Sabatier.

## Author contributions

**Conceptualization:** Celmira Laza-Vásquez, Marta Román, Suzette Delaloge, Sandrine De Montgolfier, Xavier Castells.

**Data curation:** Celmira Laza-Vásquez, Marta Román, Xavier Castells.

**Formal analysis:** Celmira Laza-Vásquez.

**Funding acquisition:** Marta Román, Xavier Castells.

**Investigation:** Celmira Laza-Vásquez, Marta Román, Suzette Delaloge, Sandrine De Montgolfier, Xavier Castells.

**Methodology:** Celmira Laza-Vásquez, Marta Román, Suzette Delaloge, Sandrine De Montgolfier, Xavier Castells.

**Project administration:** Celmira Laza-Vásquez, Marta Román, Xavier Castells.

**Writing – original draft:** Celmira Laza-Vásquez, Marta Román, Suzette Delaloge, Sandrine De Montgolfier, Xavier Castells.

**Writing – review & editing:** Celmira Laza-Vásquez, Marta Román, Suzette Delaloge, Sandrine De Montgolfier, Xavier Castells.

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
