## [Decision Letter · Decision Letter 0]

9 Apr 2025

plosone@plos.org. . . . A rebuttal letter that responds to each point raised by the academic editor and reviewer(s). You should upload this letter as a separate file labeled 'Response to Reviewers'.A marked-up copy of your manuscript that highlights changes made to the original version. You should upload this as a separate file labeled 'Revised Manuscript with Track Changes'.An unmarked version of your revised paper without tracked changes. You should upload this as a separate file labeled 'Manuscript'.

We look forward to receiving your revised manuscript.

Kind regards,

Eugenio Paci, MD

Academic Editor

PLOS ONE

Journal Requirements:

https://journals.plos.org/plosone/s/file?id=ba62/PLOSOne_formatting_sample_title_authors_affiliations.pdf....

Additional Editor Comments (if provided):

This is a very interesting and original manuscript. Qualitative research in the context of a randomized trial is still rare. I am convinced of the importance of considering participants' narratives, as the authors have done in previous studies cited in their reference list. This study is original precisely because it considered the experience of the health professionals involved in the MyPEBS study.

First, I would like to disclose my position. I am a member of the Monitoring Board of the MYPEBS study and therefore, I will ask a question about the compatibility of my AE function to the Editor of PLOS ONE. The point regarding the conflict of interest and, in any case, the inappropriateness of conducting the study with the possible limitations in data collection and analysis, is the main subject of the second reviewer’s comment. I am convinced that the reviewer is right given the text in its current form. I believe that qualitative research incorporated in an intervention study is an added value of the study, but there must be transparent and explicitly stated reference to MYPEBS from the title.

So, if the manuscript is revised in this direction, in my opinion, there is no such problem. The separation of qualitative researchers from research participants is sufficient, in my opinion, to ensure autonomy and allow for work that might otherwise be difficult to accomplish.

1) I agree with the reviewer that, given the context, the personal identity of respondents working in a single hospital should be protected more carefully. Table 1 should be eliminated, and some information about participants, in aggregate, should be presented as text.

2) It is also necessary, in agreement with the reviewer to specify what is meant by reaching theoretical saturation. From the way it is presented it may appear that there are a limited number of professionals to interview.

I consider the text to be too long and ask the authors to limit themselves to presenting their interesting results , cutting out all the sections about the prospects for future work. These, in my opinion, are conventional and in any case could be better developed in a later manuscript that takes into account the whole of the qualitative research work done on the MyPEBS study, rather than being presented as a consequence of the health practitioners' evaluations alone.

Reviewer 2 raises a central point because here the central issue is not his perception of risk (population risk) but the individual one, relative to the specific individual. On this basis, " accepting that some women, both women at higher risk and lower risk, will accept greater risk of harms for greater assurance of safety," and what is the purpose of risk stratification, what kind of information is needed to illustrate in the real and individual context the knowledge of the natural history of the tumor that allows the attribution of risk should be investigated. In the end, as the reviewer suggested, there is need to be explicit, also in a quantitative manner, with the existing evidence that conditions the individual decisions.

Reviewers' comments:

Reviewer's Responses to Questions

**Comments to the Author**

1. Is the manuscript technically sound, and do the data support the conclusions?

Reviewer #1: Yes

Reviewer #2: Partly

2. Has the statistical analysis been performed appropriately and rigorously?

Reviewer #1: N/A

Reviewer #2: No

3. Have the authors made all data underlying the findings in their manuscript fully available?

Reviewer #1: Yes

Reviewer #2: Yes

4. Is the manuscript presented in an intelligible fashion and written in standard English?

Reviewer #1: Yes

Reviewer #2: Yes

Reviewer #1: PONE-D-24-54911, Unanswered questions”: acceptability of a personalised breast cancer screening strategy in low-risk women by healthcare professionals

The authors explored the opinions and thoughts about implementation of personalized breast cancer screening in Spain through interviews with a variety of health care professionals.

In the introduction, the authors state, “Risk-stratified breast cancer screening is a promising strategy aiming to optimize the current model [ of breast cancer screening ] by enhancing the balance between the benefits and harms of early detection.”

Well, yes, but what exactly is meant by “enhancing” the balance of benefits and harms? What is missing in the introduction is a discussion of the inherent conflict in current models of risk stratification between benefit to risk balance on an individual basis and a population basis, including accepting that some women, both women at higher risk and lower risk, will accept greater risk of harms for greater assurance of safety.

As a prelude to discussing the responses from health professionals, the authors should briefly describe the trade-offs between benefit and harm in some models of risk stratification, the current state of evidence for the key risk factors that would be applied to risk stratification, and most especially, the absence of data that the major risk factors are associated with the breast cancer sojourn time. Just because a woman is less likely to develop breast cancer during a 1, 2, 4 (what happened to 3?) year interval, it does not mean that her breast cancer will also grow more slowly, so that the screening interval assigned to various risk profiles will always be shorter than the mean sojourn time. This would be first step towards being able to provide reassurance. It also should be noted that biennial screening already is not risk-stratified for pre-menopausal women, since the combination of density and somewhat faster tumor growth increases the risk of being diagnosed with an advanced breast cancer. What is missing in this introduction is the importance of setting the stage for the discussion of the professional’s opinions. What also is missing is whether the principle of risk stratification in this context is intended to achieve the principle of “equal treatment for equal risk,” i.e., that the goal of risk stratification would be to efficiently provide all women with an equal opportunity to have their breast cancer detected at an early stage. If the model favors high risk women on this principle over lower risk women (who, actually, have comparatively high lifetime risk compared with other chronic conditions), then it should not be surprising that many women would have less interest.

I would then characterize the response from the health professionals in a more quantitative manner, especially to the extent that their opinions are consistent with the prevailing evidence. The responses are interesting and I would retain the most illuminative opinion statements, but the narrative is rather long as submitted.

This is a novel study, but it should be grounded in the current evidence (this would benefit the reader), the variability in the concept of risk-stratified screening and expected outcomes on a population and individual basis—in most models, there are winners and losers—the higher costs of risk of errors in establishing and regularly updating risk, and then how this health professionals understand and feel about this concept.

Reviewer #2: Thank you for the interesting reading of your paper.

The paper poses an important research question to succeed the implementation of a new perspective of population-based screening which leave the concept of “a one size fits all program” to a complex concept of risk-stratified programs adjusted to fit the individual risk. Health professionals are important stakeholders in risk information and may have to answer questions while working in related areas.

This study planned to interview 14 healthcare professionals participating the MyPeBS project in Spain of whom 11 participated. Four in a focusgroup interview and 7 in individual interviews. Three did not participate. The healthcare professionals are not representative of the health care community because they work in breast cancer screening and are involved in MyPeBS. It would suit the paper to clearly state this in the discussion and as a limitation.

Do you use pseudonymization to present your participant?

I am ethically worried about the 11 participants being identified by colleagues, patients and people in Hospital del Mar? Breast cancer Screening is a small area of healthcare and the detailed information you provide make me picture all of them. Please, specify whether pseudonymization was used. It gets worse because all quotes are referred to a given participant number. Please, omit that from result tables. That information is not needed by the reader.

The result section is elaborate and present at detailed view into your data. However, the section is too long, and the reader is left without an overview of the important findings. I suggest the tables to be put in a supplement. Keep Table 1 an a few important quotes.

The discussion could benefit from knowledge about the general practitioner’s role in screening communication. They have an important role in many people's life because they seek help in their local health environment.

Finally your reference list is impressing. By thorough reading I find most papers published by the researchers involved in MyPeBS collaborating across institutions. Perhaps, it is dose not pose a problem however, the references are used as examples of agreement with other researchers in the discussion and that is not convincing reading. Please, specify when co-workers are sited and use more evidence from other research groups if possible. Otherwise cut down on citations.

The paper is well written and structured and easy to understand. Minor language revisions are needed.

Detailed feedback

Line 69, page 4

Personalization increase complexity. Please, specify complexity of what?

Line 72, page 4

“the potentially lower of extending” ? Do you mean acceptance. Please, correct the sentence.

Line 75, page 4

Brackets contain reasons for acceptance. Should habits be included?

Line 108, page 6

An actual status of the recruitment as of today would be relevant. The paper states that 53,142 women aged 40 to 70 y were recruited by mid-2023. Visiting the webpage as of March 28th, it is stated that 53,143 women are recruited. Please clarify whether that means nobody has been recruited since mid-2023? Or the study has stopped recruiting.

Line 132, page 9.

“The sample size was determined using the criterion of the theoretical saturation.”

How did you do that when the 14 healthcare professionals were selected up front?

Line 139, page 9

The interview script?

Do you mean a structured interview-guide? The answers were open for the participants?

Line 225, page 16

The heading is mistrust but distrust is used in Table 3, second question.

Table 3

In the text page 16, you make headings for three themes, but the table contains four views.

Section 3.4.1

This section could be shorter because it reflects the opinion of healthcare professionals attitude towards risk communication to citizens. Please specify, that they are laypeople in this professional field. Professional communicator can write a better and more effective communication strategy for implementation of risk-stratified breast cancer screening. Some details are not important.

Line 544-48, page 42

Can risk-communication be specified whether it’s SNP’s or high penetrating genes we communicate about. Studies are often divided into one or the other and the findings are not identical.

.

Reviewer #1: No

Reviewer #2: No

---

## [Author Response · Author response to Decision Letter 1]

27 Jun 2025

REF: PLOS ONE [PONE-D-24-54911]

“Unanswered questions”: acceptability of a personalised breast cancer screening strategy in low-risk women by healthcare professionals in the context MyPeBS study

The authors would like to thank the editor and reviewers for their comments, as their contributions have considerably enriched the new version. Suggested modifications have been made to the text, which are highlighted in the manuscript in red. You can find a specific answer to each comment below.

Dr. Paci, for health reasons, I will be away from work for a while. Therefore, I would appreciate it if any messages sent to me could also be forwarded to the other two authors of the article:

Xavier Castells: e-mail: xcastells@psmar.cat

Celmira Laza-Vásquez: celmira.laza@udl.cat

Communication 13 June

Please accept my apologies for not responding sooner. As I wrote to the editor in the letter responding to the reviewers on 22 May, I am on medical leave due to health issues and only today did I check the last message sent.

Therefore, I request that all communications be sent to Dr Xavier Castells at xcastells@psmar.cat. Dr Castells is the other author of correspondence.

1. Please provide additional details regarding participant consent. In the ethics statement in the Methods and online submission information, please ensure that you have specified (1) whether consent was informed and (2) what type you obtained (for instance, written or verbal, and if verbal, how it was documented and witnessed). If your study included minors, state whether you obtained consent from parents or guardians. If the need for consent was waived by the ethics committee, please include this information.

Additional information on informed consent was provided both online and in the manuscript. Online, the following was added to the ‘Ethics Statement’ section of the submission: ‘Individual written informed consent was obtained from the participants.’

In the article version, the following was added to section 2.5 Ethical considerations: Individual written informed consent was obtained from the participants." (page 9, line 186).

I hope that the delay in responding and making the requested modification will not affect the submission and publication process of the article.

Best regards,

Marta Roman

Additional Editor Comments (if provided):

This is a very interesting and original manuscript. Qualitative research in the context of a randomized trial is still rare. I am convinced of the importance of considering participants' narratives, as the authors have done in previous studies cited in their reference list. This study is original precisely because it considered the experience of the health professionals involved in the MyPEBS study.

1. I believe that qualitative research incorporated in an intervention study is an added value of the study, but there must be transparent and explicitly stated reference to MYPEBS from the title.

RESPONSE: Thank you for you suggestion. We have modified the title to refer to the MyPeBS project: “Unanswered questions”: acceptability of a personalised breast cancer screening strategy in low-risk women by healthcare professionals in the context MyPeBS study. (page 1)

So, if the manuscript is revised in this direction, in my opinion, there is no such problem. The separation of qualitative researchers from research participants is sufficient, in my opinion, to ensure autonomy and allow for work that might otherwise be difficult to accomplish.

2. I agree with the reviewer that, given the context, the personal identity of respondents working in a single hospital should be protected more carefully. Table 1 should be eliminated, and some information about participants, in aggregate, should be presented as text.

RESPONSE: Thank you very much for your suggestion. We appreciate the attention that you and reviewer 2 have given to this issue. We agree to remove Table 1 from the study participant characteristics and include the following description of participant characteristics in the text (page 7 line 140):

“Fourteen healthcare professionals participating in the MyPeBS project in Spain were invited in person and/or by email, and three did not respond. The sample included 11 professionals with an average of 15.5 years of experience in the breast cancer screening programme (range 2 to 28 years), with different professions and roles within the project: three doctors specialising in radiology and one in gynaecology and obstetrics; one doctor specialising in public health and epidemiology; four senior diagnostic imaging technicians; and two research technicians responsible for recruiting women for the study. 73.7% of the subjects were women and the average age was 40.7 years (range 25 to 65 years)”.

3. It is also necessary, in agreement with the reviewer to specify what is meant by reaching theoretical saturation. From the way it is presented it may appear that there are a limited number of professionals to interview.

RESPONSE: Thank you for you suggestion. It is evident that both you and reviewer 2 are correct in their assertions. When a predetermined number of participants is established, the criterion of information saturation does not dictate the size or configuration of the sample. However, it should be noted that, despite the limited number of participants in the study and the heterogeneity of their profiles, the information became saturated. The analysis, in conjunction with the collection of information, demonstrated theoretical saturation of all topics commencing with participant number 8.

Therefore, the statement “The sample size was determined using the criterion of theoretical saturation (17)” was removed.

However, we have made the following clarification in the text (page 7 line 150): … “and from informant number 8 onwards, no new information was obtained that would permit further expansion or deepening of the results. This finding suggests that theoretical saturation of information was achieved”.

4. I consider the text to be too long and ask the authors to limit themselves to presenting their interesting results, cutting out all the sections about the prospects for future work. These, in my opinion, are conventional and in any case could be better developed in a later manuscript that takes into account the whole of the qualitative research work done on the MyPEBS study, rather than being presented as a consequence of the health practitioners' evaluations alone.

Response: Thank you for you suggestion. We agree with the suggestion, so we have removed Theme 4. Recommendations for the implementation of personalised breast cancer screening in Spain.

We have also made the following modifications:

• Removed Table 5 “Table 5. Recommendations for the implementation of personalised breast cancer screening in Spain”.

• The objective in the abstract and the introduction was modified. It was also included that the context of the study was the MyPeBS project (page 2 line 29 and page 6 line 114): “We explored healthcare professionals involved in the MyPeBS study views on the acceptability of a personalised strategy for breast cancer screening in women deemed to be at low-risk”.

• In the abstract removed the results on the recommendations of health professionals: “To promote acceptability, they recommended "staggering" implementation, training of healthcare professionals, and development of socialization strategies.”

In the discussion, some sections on the discussion of the results of topic 4 of the results were removed:

Our results are consistent with previous approaches and, given the above-mentioned difficulties, proposes innovative solutions such as the creation of "information exchange networks" to facilitate the regular training of primary care professionals. This is a novel and undocumented recommendation in the scientific literature, where only online courses, websites and seminar-style conferences are proposed(Ayoub et al., 2023; Lapointe et al., 2022).

An issue not highlighted by the healthcare professionals participating in the study, but widely discussed in the literature, is the integration and involvement of nurses in the implementation(French et al., 2022; Lapointe et al., 2023b; Laza-Vásquez, Hernández-Leal, et al., 2022). This is a key consideration because of nurses’ skills and their contributions to the development of current screening programs(Bakhai et al., 2024). In addition, their involvement would help to "spread" the workload in primary care teams(Ayoub et al., 2023; Laza-Vásquez, Codern-Bové, et al., 2022; G. Taylor et al., 2022).

For the purpose of informing and educating women, a relevant aspect of this study is the participants' proposal to amplify "women's voices" to help the socialisation of personalisation through women-to-women dialogue. This approach is based on the "experiential knowledge" of lay women and the scientific knowledge of experts. This strategy could be valid for the socialisation of personalisation given its known benefits in raising public awareness of breast cancer through patient associations and "expert patients"(Foucaut et al., 2023; Galvin et al., 2022; McDonald et al., 2023; Rivest et al., 2020) and for survivors of the disease as a valuable source of information and support(Fong et al., 2017; Harmon et al., 2021) and by fostering a "sense of community" among women(Buki et al., 2023).

Finally, and in line with other studies, participants supported a proposal for a "staggered" implementation. Several elements support this suggestion: women already participating in screening programmes are "familiar" with this model(L. C. Taylor, Law, et al., 2023) and breaking the "promise" of biennial screening could create mistrust(Woof et al., 2021) and potentially provoke feelings of "injustice" among women(LA Dunlop et al., 2024). Conversely, less intensive screening would be more acceptable to younger women, as they are less aware of current screening intervals(Woof et al., 2021) and consider the disease to affect women older than 50 years (Hindmarch et al., 2023) and mammography to be a "cumbersome" examination(L. C. Taylor, Law, et al., 2023).

And others were modified:

- Page 18 line 404: “Therefore, to promote acceptability among women deemed to be at low risk, several recommendations made by other researchers should be taken into account:…”

- Page 19 line 438: “In order to inform, educate and socialise women about personalised screening strategies, community initiatives such as patient associations and the “expert patient” strategy could be beneficial, as they have proven to be valuable sources of information and support and promote a ‘sense of community’ among women”

5. Reviewer 2 raises a central point because here the central issue is not his perception of risk (population risk) but the individual one, relative to the specific individual. On this basis, " accepting that some women, both women at higher risk and lower risk, will accept greater risk of harms for greater assurance of safety," and what is the purpose of risk stratification, what kind of information is needed to illustrate in the real and individual context the knowledge of the natural history of the tumor that allows the attribution of risk should be investigated. In the end, as the reviewer suggested, there is need to be explicit, also in a quantitative manner, with the existing evidence that conditions the individual decisions.

RESPONSE: We appreciate the suggestion, which, as researchers, makes us reflect on the urgency of future studies that use mixed approaches and/or methodological triangulation processes to obtain more comprehensive and in-depth results. However, a qualitative approach was proposed for this study precisely because it was conducted in the context of the MyPeBS randomized trial, in which a quantitative analysis is being carried out.

Therefore, we emphasise presenting the qualitative findings in a comprehensive and in-depth manner that reflects their richness and the different points of view and perspectives from the diverse experiences and roles of the participants. Perhaps, although the narratives may seem lengthy, it is precisely these broad, in-depth narratives, in which all points of view are respected, that achieve the objective of qualitative research.

Reviewer 1 comments

The authors explored the opinions and thoughts about implementation of personalized breast cancer screening in Spain through interviews with a variety of health care professionals.

1. In the introduction, the authors state, “Risk-stratified breast cancer screening is a promising strategy aiming to optimize the current model [of breast cancer screening] by enhancing the balance between the benefits and harms of early detection.”

Well, yes, but what exactly is meant by “enhancing” the balance of benefits and harms? What is missing in the introduction is a discussion of the inherent conflict in current models of risk stratification between benefit to risk balance on an individual basis and a population basis, including accepting that some women, both women at higher risk and lower risk, will accept greater risk of harms for greater assurance of safety.

RESPONSE: Thank you very much for the suggestion. We have included information to clarify the statement.

… by increasing sensitivity in women at higher risk of breast cancer and specificity in women at lower risk, with an overall balance no worse than current screening. (Page 4 line 56)

… while decreasing screening frequency and harms such as overdiagnosis and false positive results. (Page 4 line 60)

2. As a prelude to discussing the responses from health professionals, the authors should briefly describe the trade-offs between benefit and harm in some models of risk stratification, the current state of evidence for the key risk factors that would be applied to risk stratification, and most especially, the absence of data that the major risk factors are associated with the breast cancer sojourn time. Just because a woman is less likely to develop breast cancer during a 1, 2, 4 (what happened to 3?) year interval, it does not mean that her breast cancer will also grow more slowly, so that the screening interval assigned to various risk profiles will always be shorter than the mean sojourn time. This would be first step towards being able to provide reassurance. It also should be noted that biennial screening already is not risk-stratified for pre-menopausal women, since the combination of density and somewhat faster tumor growth increases the risk of being diagnosed with an advanced breast cancer. What is missing in this introduction is the importance of setting the stage for the discussion of the professional’s opinions. What also is missing is whether the principle of risk stratification in this context is intended to achieve the principle of “equal treatment for equal risk,” i.e., that the goal of risk stratification would be to efficiently provide all women with an equal opportunity to have their breast cancer detected at an early stage. If the model favors high risk women on this principle over lower risk women (who, actually, have comparatively high lifetime risk compared with other chronic conditions), then it should not be surprising that many women would have less interest.

RESPONSE: We appreciate the suggestions made to improve the introduction to the article, with which we agree. Upon rereading, we also consider it necessary to include information on the opinions of health professionals regarding the personalised breast cancer screening model and, in particular, on their experiences with women who are considered to be at low risk. We have therefore included the following information (page 5 line 96):

“There is limited evidence on the opinions of healthcare professionals involved in personalised screening projects regarding the acceptability of this strategy. In general, opinions are positive, considering it a logical step in breast screening. They believe that this strategy will particularly benefit high-risk women by increasing the frequency of screening, reducing mortality from the disease and increasing the survival of women diagnosed. For low-risk women, they report a reduction in overdiagnosis and false positives. These advantages will in

---

## [Decision Letter · Decision Letter 1]

27 Jul 2025

Dear Dr. Román,

Thank you for submitting your manuscript to PLOS ONE. After careful consideration, we feel that it has merit but does not fully meet PLOS ONE’s publication criteria as it currently stands. Therefore, we invite you to submit a revised version of the manuscript that addresses the points raised during the review process.

We look forward to receiving your revised manuscript.

Kind regards,

Eugenio Paci, MD

Academic Editor

PLOS ONE

Journal Requirements:

Additional Editor Comments:

Thank you for the revision and changes in the title and manuscript. I think the manuscript is now more readable, and you avoid the problem of the conflict of interest and the issues about privacy.

Still, there are problems, as the reviewer underlines in its second revision, you should try to better address. These issues are at the core of the message of your paper and condition the meaning of the paper for the journal's audience.

So, I strongly suggest not avoiding the issue and trying to argue the comment of the reviewer.

There is no contrast between a population-based approach versus an individual approach. In this kind of evaluation, we always apply individual estimates based on estimates from population studies.

The issue at the core of the discussion, in my opinion, and I guess in agreement with the reviewer, is that this qualitative assessment occurs in the context of an RCT, MyPeBs, which is aimed at overcoming the limited knowledge we have about personalized risk.

I did not read any consideration from the professionals about the level of uncertainty being the reason why a woman is recruited in a trial like this.

This is particularly true on the inter-screening interval. In the personalized assessment, we can define a women as at low risk given for example a biomarker negativity.

Does that mean that lower risk means slower tumor growth?. More data are needed; this study will contribute to this, we hope. However, I should ask to be very clear about this point. Whereas sensitivity is important and personalised screening will primarily contribute to better detection, we must be very careful about communication of what we do not know both in the safety of screening in lower risk women, both about side effects, which we do not really know in the new context of risk based screening (false positive and, most, overdiagnosis)

In conclusion, I invite you to a revision considering these and the reviewer's suggestions both in the text and in the abstract.

Reviewers' comments:

Reviewer's Responses to Questions

**Comments to the Author**

Reviewer #1: (No Response)

2. Is the manuscript technically sound, and do the data support the conclusions?

Reviewer #1: No

3. Has the statistical analysis been performed appropriately and rigorously?

Reviewer #1: N/A

4. Have the authors made all data underlying the findings in their manuscript fully available?

Reviewer #1: Yes

5. Is the manuscript presented in an intelligible fashion and written in standard English?

Reviewer #1: Yes

Reviewer #1: I'm disappointed that the authors have not addressed my comments, specifically that they overlook the inherent problem that personalized screening is commonly conceptualized as improving detection of breast cancer in higher risk women, and reducing recalls in lower risk women. As they say very early in the introduction, "by increasing sensitivity in women at higher risk of breast cancer and specificity in women at lower risk, with an overall balance no worse than current screening." Is this progress? Not in my opinion. We need to do a much better job at detecting breast cancer early in all women. And we need to do this more efficiently, which means also reducing avoidable recalls. What is seriously flawed in this study is that health care professionals were interviewed about screening concepts for which they are not well informed, but demonstrated, I think, an impressive degree of caution about the potential costs of personalized screening to lower risk women. The authors should also have shown greater awareness of the limitations of our current risk models, and here is the main point, the very weak data linking risk and sojourn time. The authors seem to believe that women's concerns about being diagnosed with a worse prognosis cancer when the screening interval is lengthened can be addressed with better education about the advantages of avoiding overdiagnosis and false positives, setting aside that we do not have any estimates of breast cancer overdiagnosis that we can cite with confidence, let alone overdiagnosis associated with the common risk factors. In my review, I asked simply that the authors note that we do not have any data that demonstrates that the primary goal of screening, which is to detect the killing cancers early, can be improved for all women with a personalized model that screens higher risk women more frequently and lower risk less frequently. For some reason, the current thinking is that more screening for some women has to be balanced with less screening for other women. There is no such requirement, nor do we have the knowledge to extend screening to 3 and 4 year intervals in lower risk women without resulting in increasing their risk of being diagnosed with an advanced breast cancer. It is a worthy goal to determine how to screen some women less frequently safely. But all women deserve screening that delivers improved sensitivity and specificity. Until we can do that, the priority has to be sensitivity.

.

Reviewer #1: No

---

## [Author Response · Author response to Decision Letter 2]

9 Sep 2025

REF: PLOS ONE [PONE-D-24-54911]

“Unanswered questions”: acceptability of a personalised breast cancer screening strategy in low-risk women by healthcare professionals in the context MyPeBS study

The authors would like to thank the editor and reviewers for their comments, as their contributions have greatly enriched the new version. The suggested changes have been incorporated into the text and are highlighted in red in the manuscript. Below is the response to the editor's and reviewer's suggestions.

Additional Editor Comments:

Thank you for the revision and changes in the title and manuscript. I think the manuscript is now more readable, and you avoid the problem of the conflict of interest and the issues about privacy.

Still, there are problems, as the reviewer underlines in its second revision, you should try to better address. These issues are at the core of the message of your paper and condition the meaning of the paper for the journal's audience.

So, I strongly suggest not avoiding the issue and trying to argue the comment of the reviewer.

There is no contrast between a population-based approach versus an individual approach. In this kind of evaluation, we always apply individual estimates based on estimates from population studies.

The issue at the core of the discussion, in my opinion, and I guess in agreement with the reviewer, is that this qualitative assessment occurs in the context of an RCT, MyPeBs, which is aimed at overcoming the limited knowledge we have about personalized risk.

I did not read any consideration from the professionals about the level of uncertainty being the reason why a woman is recruited in a trial like this.

This is particularly true on the inter-screening interval. In the personalized assessment, we can define a women as at low risk given for example a biomarker negativity.

Does that mean that lower risk means slower tumor growth?. More data are needed; this study will contribute to this, we hope. However, I should ask to be very clear about this point. Whereas sensitivity is important and personalised screening will primarily contribute to better detection, we must be very careful about communication of what we do not know both in the safety of screening in lower risk women, both about side effects, which we do not really know in the new context of risk based screening (false positive and, most, overdiagnosis)

In conclusion, I invite you to a revision considering these and the reviewer's suggestions both in the text and in the abstract.

Reviewer #1

I'm disappointed that the authors have not addressed my comments, specifically that they overlook the inherent problem that personalized screening is commonly conceptualized as improving detection of breast cancer in higher risk women, and reducing recalls in lower risk women. As they say very early in the introduction, "by increasing sensitivity in women at higher risk of breast cancer and specificity in women at lower risk, with an overall balance no worse than current screening." Is this progress? Not in my opinion. We need to do a much better job at detecting breast cancer early in all women.

And we need to do this more efficiently, which means also reducing avoidable recalls. What is seriously flawed in this study is that health care professionals were interviewed about screening concepts for which they are not well informed, but demonstrated, I think, an impressive degree of caution about the potential costs of personalized screening to lower risk women. The authors should also have shown greater awareness of the limitations of our current risk models, and here is the main point, the very weak data linking risk and sojourn time.

The authors seem to believe that women's concerns about being diagnosed with a worse prognosis cancer when the screening interval is lengthened can be addressed with better education about the advantages of avoiding overdiagnosis and false positives, setting aside that we do not have any estimates of breast cancer overdiagnosis that we can cite with confidence, let alone overdiagnosis associated with the common risk factors.

In my review, I asked simply that the authors note that we do not have any data that demonstrates that the primary goal of screening, which is to detect the killing cancers early, can be improved for all women with a personalized model that screens higher risk women more frequently and lower risk less frequently. For some reason, the current thinking is that more screening for some women has to be balanced with less screening for other women. There is no such requirement, nor do we have the knowledge to extend screening to 3 and 4 year intervals in lower risk women without resulting in increasing their risk of being diagnosed with an advanced breast cancer. It is a worthy goal to determine how to screen some women less frequently safely. But all women deserve screening that delivers improved sensitivity and specificity. Until we can do that, the priority has to be sensitivity.

Response: We appreciate the suggestion from the reviewer and editor.

We agree that there is insufficient evidence to support the benefits of personalised screening compared to the current screening model. As the reviewer points out, increased screening for some women is not necessarily offset by reduced screening for low-risk women, and we cannot guarantee that there is no risk of detecting cancers with a poorer prognosis if the screening interval is increased to 3 or 4 years.

Precisely, the research carried out by the MyPeBS study aims to provide evidence on this issue. In the meantime, we start from a position of uncertainty, as the reviewer comments. However, we do start from the hypothesis that personalised screening will lead to an improvement, and this is what ethically justifies the MyPeBS study.

We would also like to point out that the qualitative study aimed to describe the opinions and acceptability of healthcare professionals regarding the reduction of screening for women who are at low risk of breast cancer. Like researchers from the BC-Predict Project in the United Kingdom, who, as in our case, have investigated this issue with healthcare professionals and women in a real context of personalised screening:

- Taylor G, McWilliams L, Woof VG, Evans DG, French DP. What are the views of three key stakeholder groups on extending the breast screening interval for low‐risk women? A secondary qualitative analysis. Health Expect. 2022;25(6):3287. 10.1111/HEX.13637

- McWilliams L, Woof VG, Donnelly LS, Howell A, Evans DG, French DP. Extending screening intervals for women at low risk of breast cancer: do they find it acceptable? 2021;21(1):637. 10.1186/s12885-021-08347-w

- Woof VG, McWilliams L, Donnelly LS, Howell A, Evans DG, Maxwell AJ, et al. Introducing a low-risk breast screening pathway into the NHS Breast Screening Programme: Views from healthcare professionals who are delivering risk-stratified screening. Women’s Health. 2021;17:17455065211009746. doi: 10.1177/17455065211009746

We are also aware that the qualitative study was conducted in a context of uncertainty regarding the evidence on the benefits of a personalised breast cancer screening strategy and that, as it is considered a promising strategy, two randomised studies are currently being conducted to test the superiority of the current screening method. Therefore, we agree with the reviewer's recommendation on the need to be explicit in clarifying this level of uncertainty in the current evidence, and we have made the following changes to the manuscript to provide greater clarity:

Abstrac (Page 2 line 27):

“Based on the relevant differences in breast cancer risk, strategies for personalizing population screening have been proposed. The present study was conducted in the context of the MyPEBS study, which seeks to improve the evidence on the benefits and risks of implementing personalized screening. Specifically, it evaluates the acceptability among professionals of the strategy of increasing the screening interval in low-risk women”.

Introduction (Page 4 line 56):

“The risk of developing breast cancer over a woman's lifetime depends on characteristics such as age, breast density, family history of breast cancer, history of benign lesions, or genetic predisposition, among other factors. The difference in risk can be substantial between two women based on these characteristics. However, population-based breast cancer screening programmes are based solely on age. It is in this context that it has been proposed to evaluate the effectiveness of screening based on a woman's individual risk [1].

There is still no evidence to suggest that personalized breast cancer screening is no worse than or potentially better than current screening in reducing breast cancer mortality [2]. Specifically, there is insufficient evidence to demonstrate that the primary goal of screening, which is to detect the killing cancers early, can be improved for all women with a personalized model that screens higher risk women more frequently and lower risk less frequently. However, it is suggested that this strategy could increase the benefits and reduce the harms of current screening; therefore, two prospective multicenter randomized controlled trials are being developed internationally to generate evidence on the effectiveness of personalized screening in reducing breast cancer-specific mortality compared to current screening (Wisdom in the US and MyPeBS in Europe and Israel) [3].

Page 5 line 82:

“One of the limitations of implementing personalised screening is ensuring similar sensitivity to the current level in low-risk women who are offered longer intervals between screenings. It is expected that additional examinations will be reduced if the screening interval is increased, but this should be done while ensuring equal sensitivity in this group of women”.

For our part, in the discussion section, we highlight this fact, which both the study participants and we as researchers consider to be of great importance:

Page 16 line 365:

“This study was conducted in the context of the MyPeBS study, which aims to improve the evidence on the effectiveness of a personalised screening strategy based on individual risk. Therefore, there is no robust evidence to show that personalised screening is better than the current strategy based solely on age. However, the hypothesis is that personalised screening will increase sensitivity in women at higher risk without compromising sensitivity in women at lower risk, for whom the screening interval is increased. In fact, the aim of this study is to analyse the opinion of healthcare professionals, assuming that this hypothesis is met, but with the uncertainty that it has not been confirmed”.

---

## [Decision Letter · Decision Letter 2]

30 Sep 2025

Dear Dr. Román,

plosone@plos.org. . . . A rebuttal letter that responds to each point raised by the academic editor and reviewer(s). You should upload this letter as a separate file labeled 'Response to Reviewers'.A marked-up copy of your manuscript that highlights changes made to the original version. You should upload this as a separate file labeled 'Revised Manuscript with Track Changes'.An unmarked version of your revised paper without tracked changes. You should upload this as a separate file labeled 'Manuscript'.

We look forward to receiving your revised manuscript.

Kind regards,

Eugenio Paci, MD

Academic Editor

PLOS ONE

Journal Requirements:

**Additional Editor Comments (if provided):**

I received the comment to the  revision and the review is favorable, but still there are  suggestions which are interesting. I ask the authors to consider the new comments and to consider them it in the revised text or writing a comment.

Reviewers' comments:

Reviewer's Responses to Questions

**Comments to the Author**

Reviewer #1: (No Response)

2. Is the manuscript technically sound, and do the data support the conclusions?

Reviewer #1: Yes

3. Has the statistical analysis been performed appropriately and rigorously?

Reviewer #1: N/A

4. Have the authors made all data underlying the findings in their manuscript fully available?

Reviewer #1: Yes

5. Is the manuscript presented in an intelligible fashion and written in standard English?

Reviewer #1: Yes

Reviewer #1: Thanks to the authors for their response to the reviewer's comments. [Note to the authors—lower-risk vs. higher risk is more appropriate in this context, and does not diminish the distinction between lower and higher risk women, or the purpose of your study. Even what you are calling a low-risk women carries a non-trivial 10-year and lifetime risk. We know very little about what explains never being diagnosed with breast cancer over the course of a lifetime, or how risk evolves in some women, and does not in others].

I appreciate the authors providing what they believe is corroborating information, but it seems the supporting references mainly are from like-minded investigators who believe that the pathway to improving the balance of benefits and harms for all women, is more sensitive screening (with whatever that would require) in higher risk women, where we also are more accepting of harms in the context of that risk, and less sensitive screening (here I’m referring to program sensitivity) in lower-risk women. Indeed, the citations include many of the same authors, so it is not surprising that the themes are similar, and key issues consistently are not addressed.

• Taylor, et al. stresses that the harms of screening low-risk women, who could experience greater harms from overdiagnosis. This is a common theme that low-risk women tend to be diagnosed with low-risk cancers. No evidence is offered for this belief, nor is the issue addressed of how low-risk, and thus lower incidence, can result in greater rates of overdiagnosis.

• McWilliams, et al. This paper offers a similar approach as the others….the pathway to less frequent screening among low-risk (sic) women is providing reassurance of precision in risk estimation and safety, and endorsement from their health care providers. As with the others, the ability to estimate risk is sufficient; correlation with risk and the aggressivity of tumors is assumed.

• Woof, et al. opens with an unproven assertion that low-risk women are more likely to be diagnosed with low-grade and in situ tumors, and that for every breast cancer death prevented, 3 cases are overdiagnosed (the UK report, and a largely meaningless statistic). They state, “it is argued by some that the benefits of screening outweigh the harms.” Really? Argued by some? And, this side of the argument is balanced by a letter to the editor by Donzelli following the UK Independent Review stressing that breast cancer screening does not reduce all-cause mortality. Who knew?

In these three papers, the issue of numeracy (what does it mean to have a 2% risk over 10 years?...it means 1 in 50 women will be diagnosed with breast cancer) and how well do clinicians understand the issues of sojourn time, tumor characteristics, and absolute risk (not RR) and the lack parallel evidence to answer these question for the lower risk group vs. the higher risk group.

On page 2, line 27, the authors highlight revised text that states that the present study was conducted I the context of the MyPEBS study, “which seeks to improve the evidence on the benefits and risks of implementing personalized screening. Specifically, it evaluates the acceptability among professionals of the strategy of increasing the screening interval in low-risk women”.

Of course this is important, but we know that the average health professional is as uninformed about these issues as their patients. The authors seem to be putting the cart before the horse. The evidence that lower-risk women can be screened safely at wider intervals requires screening outcome data before there can be a public health campaign to influence clinicians and the target population. It is quite ok to plan in advance for that campaign, in fact it is the responsible thing to do, what the planning should consider all outcomes, not only for the outcomes they hope to observe.

Change to the introduction, page 4, line 56. I agree with this narrative. No question, personalized screening is where we need to be vs. one-size-fits-all.

Page 5, line 82: “…It is expected that additional examinations will be reduced if the screening interval is increased, but this should be done while ensuring equal sensitivity in this group of women”. Good. But, suggest that the authors say “equal sensitivity in the detection of early-stage breast cancers by screening in this group of women.” Sensitivity may be minimally diminished, but tumor characteristics could be worse. In other words, in addition to longer term outcomes, you would want to know that the tumor characteristics at diagnosis were no worse, the interval cancer rate was no worse, the risk of lymphedema through aggressive node dissection was no worse, and the need for toxic therapies was no worse.

Page 16 line 365: I have had a look at the MyPeBS website and have reviewed the background materials. It is rather remarkable that this study puts so much faith in risk estimation as offering a sweeping opportunity to improve outcomes, when the only thing that is reasonably certain is there are likely to be better outcomes from identifying a higher-risk group (including women at higher-risk that were not known to be), and screening them more aggressively. The opportunity to avoid the harms of screening in a lower-risk group focuses on reducing the opportunity for harms by reducing the exposure to screening, which includes the diagnosis of cancers that have a higher-risk of being overdiagnosed, for which there is zero evidence of that potential, or to be treated less aggressively, for which all women, lower and higher risk, might benefit if the cancer is localized and low grade.

Page 18, Line 411: The authors write, “Two European studies suggest that fear levels are high, even among women younger than 50 years, prompting them to undergo screening mammograms, even when not medically indicated and despite a high level of knowledge about the disease [22,23].” I think it would be less ideological to replace “even when not medically indicated,” with something that simply stresses in these countries screening before age 50 was not offered by the national service. There are numerous countries that offer screening to women in their 40s, and the evidence shows that screening is effective in this group. Saying it is not medically indicated implies that absolute risk is too low, or that screening is ineffective. Data from England and the US have shown clearly that there are only small differences in the burden of disease among women ages 45-49 and 50-54. Even if the country does not offer screening before age 50, it is not unreasonable for women to want to begin screening earlier. And, for some women, personalized screening would recommend that.

Page 20, line 451: Another essential and pressing issue raised in the study [is the training of primary care professionals [is the need to train primary care professionals], given the decisive role they will play in implementing a personalised screening model [30]. In this sentence, the need for training is stressed twice….I think the authors changed the way they wished to express this, and neglected to delete the text they wished to replace.

General comment: One worrisome aspect of MyPeBS, and the narrative of this paper, is what feels like a high degree of confidence the personized screening will be shown to be effective for both higher and lower risk women, in effect that the interventions being tested will be shown to be efficacious for both groups. Although the authors state early on that noninferiority has not been demonstrated, and that this is the purpose of myPeBS, there seems to be little doubt about the outcome and the need to prepare for it. I agree that too often there is no preparation for the outcome of a large trial, and this delays implementation, but at this time, there should be some acknowledgement that the implementation challenges are uncertain overall, and that with respect to screening intervals, they may be shortened for higher-risk women, but not for lower-risk women. On line 434, the authors state, “Therefore, to promote acceptability among women deemed to be at low risk, several recommendations made by other researchers should be considered:…” Where there are statements like, this, I would encourage the authors to simply acknowledge, “if the findings from MyPeBS support wider screening intervals for lower-risk women, then……” Otherwise, it appears that moving ahead involves persuading women to accept a wider screening interval as a desirable tradeoff to reduce harms, in other words, taking a chance that safety will not be compromised in their life vs. the greater certainty of recalls (with the usually cited false positives, biopsies, and overdiagnosis) with more frequent screening, becomes the implementation challenge. Not only would that be debatable, but because the study is still underway, we’re not there yet, and it conveys that this would be the plan whatever the outcome.

If it becomes evident that lower-risk women can be screened safely at wider intervals, who among us wouldn’t want that? And, we (scientists and clinicians) would face a challenge persuading women that this is not about cost saving, but rather better outcomes. But, the final results of the study are not in. This study has identified what the investigators will face depending on the results. I encourage the authors to be clear that the study outlines the concerns of health professionals and what they perceive as their patients’ concerns, which will need to be taken into consideration depending on MyPeBS outcomes.

.

Reviewer #1: No

While revising your submission, please upload your figure files to the Preflight Analysis and Conversion Engine (PACE) digital diagnostic tool, https://pacev2.apexcovantage.com/. PACE helps ensure that figures meet PLOS requirements. To use PACE, you must first register as a user. Registration is free. Then, login and navigate to the UPLOAD tab, where you will find detailed instructions on how to use the tool. If you encounter any issues or have any questions when using PACE, please email PLOS at . PACE helps ensure that figures meet PLOS requirements. To use PACE, you must first register as a user. Registration is free. Then, login and navigate to the UPLOAD tab, where you will find detailed instructions on how to use the tool. If you encounter any issues or have any questions when using PACE, please email PLOS at . PACE helps ensure that figures meet PLOS requirements. To use PACE, you must first register as a user. Registration is free. Then, login and navigate to the UPLOAD tab, where you will find detailed instructions on how to use the tool. If you encounter any issues or have any questions when using PACE, please email PLOS at . PACE helps ensure that figures meet PLOS requirements. To use PACE, you must first register as a user. Registration is free. Then, login and navigate to the UPLOAD tab, where you will find detailed instructions on how to use the tool. If you encounter any issues or have any questions when using PACE, please email PLOS at figures@plos.org. Please note that Supporting Information files do not need this step.. Please note that Supporting Information files do not need this step.

---

## [Author Response · Author response to Decision Letter 3]

17 Mar 2026

We have carefully review and considered the reviewers comments. We provide a Point by Point response letter to reviewers’ comments, as well as a clean and modified version of the manuscript with track changes for consideration.

We have accommodated the editor’s and reviewer’s suggestions throughout the manuscript, mostly in the Introduction and Discussion sections. We have placed emphasis in the assessment and monitoring of lower risk women, and the need to monitor the sensitivity and the detection of early stage breast cancers in this group of women if the screening interval is increased.

We believe it is important not to lose sight of the purpose of the article. As it is a qualitative study on perspectives and opinions of Health Care Professionals. It is out of the Scope of this paper to propose recommendations or quantify how to monitor personalized screening. The study offers relevant insights into the factors influencing the acceptability of decreasing the frequency of screening in lower-risk women, highlighting health professional’s recommendations on how to promote acceptability in this group of women. We hope that the modifications made are satisfactory to the editor and reviewers, and that they have helped to improve the scientific quality of this study.

---

## [Decision Letter · Decision Letter 3]

26 Mar 2026

“Unanswered questions”: acceptability of a personalised breast cancer screening strategy in lower-risk women by healthcare professionals in the context of the MyPeBS study

PONE-D-24-54911R3

Dear Dr. Román,

We’re pleased to inform you that your manuscript has been judged scientifically suitable for publication and will be formally accepted for publication once it meets all outstanding technical requirements.

Kind regards,

Eugenio Paci, MD

Academic Editor

PLOS One

Additional Editor Comments (optional):

As the reviewer, I appreciated the work done and the improvements and clarification in the text. In my view, the manuscript shows how important qualitative research can be and the need for the involvement of health professionals and women at risk in the debate on personalized screening and identification of a lower risk group.

Reviewers' comments:

Reviewer's Responses to Questions

**Comments to the Author**

Reviewer #1: All comments have been addressed

2. Is the manuscript technically sound, and do the data support the conclusions?

Reviewer #1: Yes

3. Has the statistical analysis been performed appropriately and rigorously?

Reviewer #1: N/A

4. Have the authors made all data underlying the findings in their manuscript fully available?

Reviewer #1: Yes

5. Is the manuscript presented in an intelligible fashion and written in standard English?

Reviewer #1: Yes

Reviewer #1: I want to thank the authors for their patience and for their willingness to address my comments about what is known and not known at this time regarding risk stratification. We all agree that breast cancer screening should strive to maximize benefits and minimize harms, and there definitely is an inherent challenge to meet this goal at both the individual and population level. Wishing you the best for your study.

.

Reviewer #1: No

---

## [Editor Report · Acceptance letter]

PONE-D-24-54911R3

PLOS One

Dear Dr. Román,

I'm pleased to inform you that your manuscript has been deemed suitable for publication in PLOS One. Congratulations! Your manuscript is now being handed over to our production team.

Kind regards,

on behalf of

Dr. Eugenio Paci

Academic Editor

PLOS One